# OPEN-WORLD SEMI-SUPERVISED LEARNING

## ABSTRACT

Supervised and semi-supervised learning methods have been traditionally designed for the closed-world setting which is based on the assumption that unlabeled test data contains only classes previously encountered in the labeled training data. However, the real world is often open and dynamic, and thus novel previously unseen classes may appear in the test data or during the model deployment. Here, we introduce a new open-world semi-supervised learning setting in which the model is required to recognize previously seen classes, as well as to discover novel classes never seen in the labeled dataset. To tackle the problem, we propose ORCA, an approach that jointly learns a feature representation and a classifier on the labeled and unlabeled subsets of the data. The key idea in ORCA is in introducing uncertainty based adaptive margin that effectively circumvents the bias caused by the imbalance of variance between seen and novel classes. We demonstrate that ORCA accurately discovers novel classes and assigns samples to previously seen classes on standard benchmark image classification datasets, including CIFAR and ImageNet. Remarkably, despite solving the harder task ORCA outperforms semi-supervised methods on seen classes, as well as novel class discovery methods on unseen classes, achieving $7\%$ and $151\%$ improvements on seen and unseen classes of the ImageNet dataset.

## 1 INTRODUCTION

With the advent of deep learning, remarkable breakthroughs have been achieved and current machine learning systems excel on tasks with large quantities of labeled data. Despite the strengths, the vast majority of models are designed for the closed-world setting rooted in the assumption that training and test data come from the same set of predefined classes. This assumption, however, rarely holds in practice, as labeling data depends on the domain-specific knowledge which can be severely incomplete and insufficient to account for all possible scenarios. Thus, it is unrealistic to expect that one can identify and prelabel all categories/classes ahead of time, and manually supervise machine learning models.

In contrast to the commonly assumed closed world, the real world is inherently dynamic and open — new classes can emerge in the test data that have never been encountered during training. Open-world setting requires the models to be able to classify previously seen classes, but also effectively handle never-before-seen classes. This task is very natural to human intelligence; children can effortlessly recognize previously learnt concepts, but also detect the patterns and differences of the new ones. However, it is still an open question whether we can design versatile models that can successfully deal with the world of unknown, while not forgetting the world of known.

Semi-supervised learning (SSL) (Chapelle et al., 2009) aims in leveraging unlabeled data when labels are difficult and costly to obtain. Recent works (Oliver et al., 2018; Chen et al., 2020b) show that incorporating novel classes in the unlabeled set degrades performance of SSL methods. To alleviate this limitation, Guo et al. (2020) ensure safety of SSL with the presence of novel classes as well. However, the ability to differentiate between seen and unseen classes is not sufficient as we need methods that can properly handle unseen classes. On the other hand, methods for discovering novel classes (Hsu et al., 2018; 2019; Han et al., 2019; 2020) utilize labeled data solely to learn a richer representation, and are not able to recognize seen and discover unseen classes at the same time.

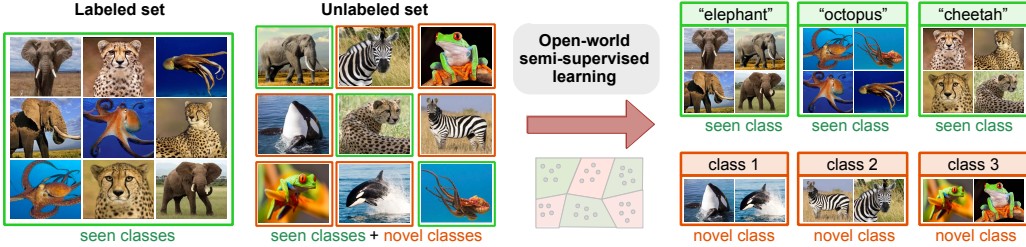

Figure 1: In the open-world semi-supervised learning, the unlabeled dataset may contain classes that have never been encountered in the labeled set. The model needs to be able to classify samples into previously seen classes, but also distinguish between unseen classes.

Here, we introduce open-world semi-supervised learning. In this setting, the unlabeled dataset may contain classes that have never been seen in the labeled dataset, and the model needs to be able to: (i) recognize when a sample from the unlabeled data belongs to one of the seen classes present in the labeled dataset, and (ii) automatically discover novel/unseen classes without any previous knowledge (Figure 1). The latter requires the ability to identify new features that can separate unseen classes. Open-world SSL is related, but differs from, continual learning (Kirkpatrick et al., 2017), generalized zero-shot learning (Xian et al., 2017), open-set (Scheirer et al., 2012) and open-world recognition (Bendale & Boult, 2015). In particular, by utilizing both labeled and unlabeled data open-world SSL relies on the transductive inference unlike the continual learning which sequentially learns new tasks while trying to mitigate catastrophic forgetting. In contrast to the generalized zero-shot learning, open-world SSL does not assume prior knowledge about the unseen classes. Finally, opposed to the open-set and open-world recognition, it requires separation of unknown classes and does not need any external supervision, respectively.

To address the challenges of open-world SSL, we propose ORCA (open-world with uncertainty based adaptive margin), an approach that can discover novel classes, while at the same time achieves high performance on classifying previously seen classes. Using both labeled and unlabeled data, ORCA learns a joint embedding function parameterized by a convolutional neural network, and linear classifier that assigns samples to previously seen classes, or to novel classes discovered by ORCA. The starting point is to initialize the model using self-supervised pretraining which previously showed effectiveness for semi-supervised learning (Zhai et al., 2019) and novel class discovery (Han et al., 2020; Van Gansbeke et al., 2020). The objective function in ORCA consists of three main components: (i) supervised loss on labeled data, (ii) pairwise loss on labeled and unlabeled data estimated using pseudo-labels inferred from the most confident pairwise similarities, and (iii) regularization that prevents the model to assign all unlabeled samples to the same class. However, naively combining supervised and pairwise losses leads to the bias towards seen classes which reduces the ability to adapt to novel classes. To mitigate the bias, the key idea in ORCA lies in introducing uncertainty based adaptive margin in the supervised loss that gradually decreases plasticity and increases discriminability of the model during training. We evaluate ORCA on benchmark image classification datasets. The results show that ORCA significantly outperforms SSL methods on the task of recognizing previously seen classes, as well as novel class discovery methods on the task of discovering unseen classes. On the latter task, ORCA improves performance of baseline methods by $51\%$ on CIFAR-100 and $151\%$ on ImageNet-100 dataset.

## 2 RELATED WORK

Open-world SSL lies on the intersection of semi-supervised learning, novel class discovery and open-world recognition.

**Semi-supervised learning (SSL).** While the literature on SSL (Chapelle et al., 2009) is vast, two most explored directions are to utilize the structure of the unlabeled data using consistency regularization (Sajjadi et al., 2016; Laine & Aila, 2016), or entropy minimization (Grandvalet & Bengio, 2005). Closely related to our work are pseudo-labeling based approaches (Lee, 2013; Sohn et al., 2020) which generate pseudo-labels for more confident unlabeled samples and use them as targets in a standard supervised loss function. Under the typically assumed closed-world assumption, SSL methods achieve highly competitive performance to supervised methods; however, recent works

(Oliver et al., 2018; Chen et al., 2020b) show that including novel classes in the unlabeled set can hurt the performance compared to not using any unlabeled data. To mitigate the negative effects, DS$^3$L (Guo et al., 2020) aims in assigning low weights to samples from novel classes. Yet, rejecting samples from novel classes is usually not enough (Boult et al., 2019). Thus, open-world SSL aims in solving more realistic and challenging task, requiring from the model to discover novel classes in the unlabeled data and group them into semantically meaningful clusters.

**Novel class discovery.** Novel class discovery, often referred to as cross-task transfer learning (Hsu et al., 2018), is a recently tackled problem related to deep learning based clustering methods (Xie et al., 2016; Yang et al., 2016; 2017; Chang et al., 2017). In contrast to clustering, novel class discovery assumes prior knowledge given in the form of labeled dataset. The task is then to cluster unlabeled dataset consisting of similar, but completely disjoint, classes than those present in the labeled dataset. The main idea is to leverage knowledge of the known classes to improve representation learning on the novel classes. Hsu et al. (2018; 2019) propose to transfer predictive pairwise similarities from labeled to unlabeled data by posing the categorization problem as a surrogate same-task problem. Deep Transfer Clustering (Han et al., 2019) extends the deep clustering framework by incorporating information about the known classes. Han et al. (2020) train the model by generating pseudo-labels of the unlabeled data using rank statistics. All the aforementioned methods maintain different classifiers to generate class/cluster assignments for labeled and unlabeled datasets, which is not applicable in the more difficult open-world SSL that requires the ability to recognize seen and dicover novel classes simultaneously.

**Open-set and open-world recognition.** Open-set recognition (Scheirer et al., 2012; Geng et al., 2020) considers the scenario in which novel classes can appear during testing, and the model needs to recognize and reject samples of novel classes. Many methods have been proposed to tackle the task such as SVM-based (Scheirer et al., 2012; Jain et al., 2014), distance-based (Júnior et al., 2017), and deep learning based methods (Bendale & Boult, 2016). On the other hand, open-world recognition (Bendale & Boult, 2015; Boult et al., 2019) requires the system to incrementally learn and extend the set of known classes with novel classes. Bendale & Boult (2015) gradually label novel classes by human-in-the-loop. Open-world SSL is related to open-world recognition, but leverages unlabeled data in the learning stage and does not need any manual input.

**Margin loss.** Based on the observation that margin term in cross-entropy loss can adjust intra- and inter-class variations, losses like large-margin softmax (Liu et al., 2016), angular softmax (Liu et al., 2017), and additive margin softmax (Wang et al., 2018) have been proposed to increase the inter-class margin to achieve better classification accuracy. Cao et al. (2019) assigned different margins to different classes to encourage the optimal trade-off in generalization between frequent and rare classes. Liu et al. (2020) found that using negative margin can help to enlarge intra-class variance and reduce inter-class variance, leading to the better performance on novel classes in the few-shot learning.

## 3 METHOD

### 3.1 PROBLEM SETTING

We first formally introduce open-world semi-supervised learning. We assume that a labeled part of the dataset $\mathcal{D}_l = \{(x_i, y_i)\}_{i=1}^{m}$ consisting of $m$ samples with labels, and an unlabeled part of the dataset $\mathcal{D}_u = \{(x_i)\}_{i=1}^{n}$ consisting of $n$ unlabeled samples, are provided during the training phase. We denote the set of ground-truth classes in the labeled data and unlabeled data as $\mathcal{C}_l$ and $\mathcal{C}_u$, respectively. Novel class discovery assumes that sets of classes in labeled and unlabeled data are disjoint, *i.e.,* $\mathcal{C}_l \cap \mathcal{C}_u = \emptyset$, while (closed-world) SSL by definition assumes the same set of classes in labeled and unlabeled data, *i.e.,* $\mathcal{C}_l = \mathcal{C}_u$. In contrast, the open-world SSL, we assume $\mathcal{C}_l \cap \mathcal{C}_u \neq \emptyset$ and $\mathcal{C}_l \neq \mathcal{C}_u$. We consider $\mathcal{C}_s = \mathcal{C}_l \cap \mathcal{C}_u$ as seen classes, and $\mathcal{C}_n = \mathcal{C}_u \backslash \mathcal{C}_l$ as novel/unseen classes. Given an unlabeled example, open-world SSL requires the algorithm to either (i) correctly classify it as one of the seen classes $\mathcal{C}_s$, or (ii) group it with similar samples from unlabeled data to form one of the novel classes $\mathcal{C}_n$.

We propose an approach, named ORCA, that effectively addresses the challenges of open-world SSL. We find that the key challenge is to mitigate the bias towards seen classes caused by learning

discriminative representations faster on the seen classes compared to the novel classes, leading to the reduced quality of the estimated pseudo-labels. To circumvent this problem, we introduce uncertainty based adaptive margin in the cross-entropy loss that trades of intra-class and inter-class variance of the seen classes. The training proceeds in two phases. First, we pretrain the network using self-supervised pretraining. Then, we optimize our proposed objective function consisting of three main components, detailed in the following sections.

## 3.2 SELF-SUPERVISED PRETRAINING

In the open-world SSL, we need to jointly solve classification and clustering tasks. Solving clustering task using deep neural networks is a challenging problem that requires learning representation and clustering assignments, resulting in high sensitivity on the network initialization. Previous works on deep clustering and novel class discovery pretrained the network using autoencoder (Xie et al., 2016; Guo et al., 2017), or self-supervised learning (Han et al., 2020; Van Gansbeke et al., 2020). Pretraining step defines a prior for the parameter space which provides better initial representations compared to the random initialization. Here, we find that pretraining step is of essential importance in open-world SSL.

Therefore, to obtain more robust representations that can be used to generalize to unseen tasks, we first pretrain ORCA using self-supervised learning. Self-supervised learning formulates a pretext/auxiliary task that does not need any manual curation and can be readily applied to both labeled and unlabeled data, such as predicting patch context (Doersch et al., 2015) or image rotation (Gidaris et al., 2018). Pretext task guides the model towards learning meaningful representations in a fully unsupervised way. In particular, we rely on the SimCLR (Chen et al., 2020b) approach. We pretrain the backbone $f_\theta$ on the whole dataset $\mathcal{D}_l \cup \mathcal{D}_u$ with a pretext objective. Learned representations are then used to initialize the network for a subsequent task.

## 3.3 RECOGNIZING SEEN AND DISCOVERING NOVEL CLASSES

In our framework, we propose the objective function that jointly solves supervised classification and unsupervised clustering task. Given labeled samples $\mathcal{X}_l = \{x_i \in \mathbb{R}^N\}_i^n$ and unlabeled samples $\mathcal{X}_u = \{x_i \in \mathbb{R}^N\}_i^m$, we first apply the embedding function $f_\theta : \mathbb{R}^N \to \mathbb{R}^D$, pretrained using self-supervised learning, to obtain the feature representations $\mathcal{Z}_l = \{z_i \in \mathbb{R}^D\}_i^n$ and $\mathcal{Z}_u = \{z_i \in \mathbb{R}^D\}_i^m$ for labeled and unlabeled data, respectively. Here, $z_i = f_\theta(x_i)$ for every sample $x_i \in \mathcal{X}_l \cup \mathcal{X}_u$. On top of the pretrained network, we add a classification head consisting of a single linear layer parameterized by a weight matrix $W : \mathbb{R}^D \to \mathbb{R}^{|\mathcal{C}_l|+|\mathcal{C}_u|}$, and followed by a softmax layer. During training, we freeze the first layers of the backbone $f_\theta$ and update its last layers and classifier $W$. The final class/cluster prediction is calculated as $c_i = \text{argmax}(W^T \cdot z_i) \in \mathbb{R}$. Note that if $c_i \notin \mathcal{C}_l$, then $x_i$ belongs to novel classes. ORCA's objective consists of three components: (i) pairwise loss, (ii) supervised loss, and (iii) regularization towards uniform distribution. We assume that the number of novel classes $|\mathcal{C}_u|$ is known and given as an input to the algorithm. This is a typical assumption of clustering and novel class discovery methods. However, if it is not the case then the number of classes needs to be estimated. We address this question in the experiments.

### 3.3.1 PAIRWISE LOSS

Inspired by (Chang et al., 2017; Hsu et al., 2018), we transform the cluster learning problem into a pairwise similarity prediction task. Given the labeled dataset $\mathcal{X}_l$ and unlabeled dataset $\mathcal{X}_u$, we aim to fine-tune our embedding function $f_\theta$ and learn a similarity prediction function parameterized by a linear classifier $W$ such the samples from the same class are grouped together. To achieve this, we rely on the ground-truth annotations from the labeled set and pseudo-labels for the unlabeled set. Specifically, for the labeled set we already know which pairs should belong to the same class so we can use ground-truth labels. To obtain the pseudo-labels for the unlabeled set, we calculate the cosine distance between all pairs of feature representations $z_i$ in a mini-batch. We then rank the distances and for each sample generate the pseudo-label for its most similar neighbor. Therefore, we only generate pseudo-labels from the most confident positive pairs for each sample within the mini-batch. For feature representations $\mathcal{Z}_l \cup \mathcal{Z}_u$ in a mini-batch, we denote its closest set as $\mathcal{Z}_l' \cup \mathcal{Z}_u'$. Note that $\mathcal{Z}_l'$ is always correct since it is generated using the ground-truth labels. Pairwise loss in

ORCA is defined as a modified form of binary cross-entropy loss (BCE):

$$\mathcal{L}_{\text{BCE}} = \frac{1}{m+n} \sum_{z_i, z_i' \in (\mathcal{Z}_l \cup \mathcal{Z}_u, \mathcal{Z}_l' \cup \mathcal{Z}_u')} -\log \langle \text{softmax}(W^T \cdot z_i), \text{softmax}(W^T \cdot z_i') \rangle, \qquad (1)$$

where softmax function assigns examples to one of the seen or novel classes. For labeled examples, we have the ground truth annotations, so we use them to compute the loss. For unlabeled examples, we compute the loss based on the generated pseudo-labels. The reason behind considering only most confident pairs to generate pseudo-labels is that we find that the increased noise in pseudo-labels is detrimental to cluster learning. Further, unlike (Chang et al., 2017; Hsu et al., 2018; Han et al., 2020) we consider only positive pairs. We find that including negative pairs in our loss does not benefit learning since the majority of negative pairs can be easily recognized. Our pairwise objective with only positive pairs is similar to SCAN (Van Gansbeke et al., 2020). However, we update distances and positive pairs in an online version in order to benefit from the improved feature representation during training. On the other hand, SCAN updates only weights of the linear classifier while freezing feature representation.

### 3.4 SUPERVISED LOSS WITH UNCERTAINTY BASED ADAPTIVE MARGIN

For the supervised loss, we utilize the categorical annotations for the labeled data $\{y_i\}_{i=1}^{n}$ and optimize weights $W$ and backbone $\theta$. First, we propose the baseline for open-world SSL using the standard cross-entropy (CE) loss:

$$\mathcal{L}_{\text{CE}}^{(B)} = \frac{1}{m} \sum_{z_i \in \mathcal{Z}_l} -\log \frac{e^{W_{y_i}^T \cdot z_i}}{e^{W_{y_i}^T \cdot z_i} + \sum_{j \neq i} e^{W_{y_j}^T \cdot z_i}}. \qquad (2)$$

However, using standard cross-entropy on labeled data creates an imbalance problem between the labeled and unlabeled, *i.e.,* gradient is updated for seen classes $\mathcal{C}_s$, but not for novel classes $\mathcal{C}_n$. This can result in learning a classifier with larger magnitudes (Kang et al., 2019) for seen classes, leading the whole model to be biased towards the seen classes. To overcome the problem, in ORCA we introduce uncertainty based adaptive margin and propose to normalize logits.

**Uncertainty based adaptive margin.** Seen classes are learned faster due to the cross-entropy loss, and consequently they tend to have a smaller intra-class variance compared to the novel classes. Since the pairwise loss generates pseudo-labels by ranking distances in the feature space, the imbalance of intra-class variances among classes will result in error-prone pseudo-labels, *i.e.,* samples from novel classes will be assigned to seen classes. To mitigate this bias, we propose to use adaptive margin to reduce the gap between intra-class variance of the seen and novel classes. Intuitively, at the beginning of the training, we want to enforce a larger negative margin to encourage a similarly large intra-class variance of the seen classes with respect to the novel classes. Close to the end of training when clusters have been formed for the novel classes, we adjust the margin term to be close to 0 so that useful label information can be fully exploited by the model.

**Logits normalization.** The unconstrained magnitudes of a classifier can negatively affect the tuning of the margin. To avoid the problem, we normalize the inputs and weights of the linear classifier, *i.e.,* $z_i = \frac{z_i}{|z_i|}$ and $W_j = \frac{W_j}{|W_j|}$. We introduce an additional scaling parameter $s$ that controls the temperature of the cross-entropy loss. The design is similar the AM-Softmax (Wang et al., 2018).

Finally, supervised loss in ORCA with uncertainty based adaptive margin is defined as follows:

$$\mathcal{L}_{\text{CE}} = \frac{1}{m} \sum_{z_i \in \mathcal{Z}_l} -\log \frac{e^{s(W_{y_i}^T \cdot z_i + \lambda \bar{u})}}{e^{s(W_{y_i}^T \cdot z_i + \lambda \bar{u})} + \sum_{j \neq i} e^{s W_{y_j}^T \cdot z_i}}, \qquad (3)$$

where $\bar{u}$ is uncertainty and $\lambda$ is a regularizer defining its strength. We set $\lambda$ to 1 in all our experiments.

**Uncertainty estimation.** We propose to capture intra-class variance using uncertainty estimated from the confidence of unlabeled samples computed from the output of the softmax function. In the

binary setting, $\bar{u} = \frac{1}{|\mathcal{D}_u|} \sum_{x \in \mathcal{D}_u} \text{Var}(Y|X = x) = \frac{1}{|\mathcal{D}_u|} \sum_{x \in \mathcal{D}_u} \Pr(Y = 1|X) \cdot \Pr(Y = 0|X)$, which can be further approximated by:

$$\bar{u} = \frac{1}{|\mathcal{D}_u|} \sum_{x_i \in \mathcal{D}_u} 1 - \max_k \Pr(Y = k|X = x_i), \tag{4}$$

up to a factor of at most 2. We use the same formula as an approximation for the group uncertainty in multi-class setting.

### 3.5 REGULARIZATION TOWARDS UNIFORM DISTRIBUTION

By using only pairwise loss on the unlabeled data, ORCA could degenerate to a trivial solution of assigning all samples to the same class, *i.e.,* $|\mathcal{C}_u| = 1$. To avoid the problem, we introduce Kullback-Leibler (KL) divergence term that regularizes $\Pr(y|x \in \mathcal{D}_l \cup \mathcal{D}_l)$ to be close to a uniform distribution $\mathcal{U}$:

$$\mathcal{L}_{\text{R}} = KL(\frac{1}{m+n} \sum_{z_i \in \mathcal{Z}_l \cup \mathcal{Z}_u} \text{softmax}(sW^T \cdot z_i) \| \mathcal{U}(y)). \tag{5}$$

This term corresponds to the minimum entropy regularization used in SSL (Grandvalet & Bengio, 2005; Lee, 2013) to prevent the class distribution from being too flat. We define it more generally using KL-divergence, so that if the prior over the classes is known, we can use it instead of the uniform distribution.

### 3.6 FINAL OBJECTIVE

The final objective function used in ORCA combines pairwise binary cross-entropy loss, supervised cross-entropy loss with adaptive margin, and proposed regularization:

$$\mathcal{L} = \mathcal{L}_{\text{BCE}} + \eta_1 \mathcal{L}_{\text{CE}} + \eta_2 \mathcal{L}_{\text{R}}, \tag{6}$$

where $\eta_1$ and $\eta_2$ are regularization parameters set to 1 in all our experiments. The pseudo-code of the algorithm is summarized in Algorithm 1 in Appendix A.

## 4 EXPERIMENTS

### 4.1 EXPERIMENTAL SETUP

**Datasets.** We evaluate ORCA on standard benchmark image classification datasets: CIFAR-10, CIFAR-100 (Krizhevsky, 2009) and ImageNet (Russakovsky et al., 2015) with controllable ratios of unlabeled data and novel classes. Since the full ImageNet dataset is very large, we sub-sample 100 classes and conduct all experiments on this subset. For each dataset, we always use the first $k$ classes as seen classes, and the rest as novel classes. We label $50\%$ samples of the seen classes, and use the rest as unlabeled set.

**Evaluation metrics.** To measure performance on unlabeled data, we follow the evaluation protocol in novel class discovery (Han et al., 2019; 2020). On seen classes, we report accuracy. On novel classes, we report both accuracy and normalized mutual information (NMI). To compute accuracy on the novel classes, we first solve optimal assignment problem using Hungarian algorithm (Kuhn, 1955). When reporting accuracy on all classes jointly, we solve optimal assignment using both seen and novel classes.

**Baselines.** ORCA is a unique method that can solve both tasks defined in open-world SSL. We compare its performance on seen classes with SSL baselines, and with novel class discovery baselines on novel classes. In particular, we compare performance on seen classes with two representative SSL methods: pseudo-labeling (Lee, 2013) and DS$^3$L (Guo et al., 2020). DS$^3$L adaptively assigns

low-weights to samples from unseen classes. On unseen classes, we compare ORCA to two recently proposed novel class discovery methods: DTC (Han et al., 2019) and RankStats (Han et al., 2020). Since RankStats relies on the self-supervised pretraining but originally pretrains the network using RotNet (Gidaris et al., 2018), we pretrained RankStats using SimCLR (Chen et al., 2020a) to ensure that the differences in the performance between ORCA and RankStats are not caused by different pretraining strategy. It is important to note that both SSL and novel class discovery methods solve easier tasks compared to ORCA since they only need to recognize seen, or discover novel classes. We further include comparison to open-world SSL baseline proposed in our work in which adaptive margin cross-entropy in ORCA is replaced with standard cross-entropy loss defined in (2).

Please refer to Appendix B for implementation details. We will made code publicly available.

## 4.2 RESULTS

**Evaluation on benchmark datasets.** We report results on CIFAR-10, CIFAR-100 and ImageNet-100 datasets in Tables 1 and 2. Despite solving harder tasks compared to SSL and novel class discovery methods, ORCA outperforms all baselines on all three datasets, achieving remarkable improvements on novel class discovery task. In particular, on seen classes ORCA consistently outperforms SSL methods, outperforming DS$^3$L by 1–7%. Novel class discovery methods can not recognize seen classes, that is match classes in unlabeled dataset to previously seen classes from the labeled dataset. However, it is possible to evaluate their performance on seen classes by regarding seen classes in the same way as novel classes (denoted by star in the tables). By comparing the performance of novel class discovery methods to SSL methods and ORCA, we find that they lag far behind. On novel classes, ORCA achieves improvements over best novel class discovery baseline of 12% and 51% on CIFAR-10 and CIFAR-100 respectively, and 151% and 142% on two splits of ImageNet-100 dataset. These remarkable improvements show that ORCA is not only unique method in its ability to solve both tasks, but also outperforms other baselines on their respective tasks. Furthermore, comparison of ORCA to the proposed open-world SSL baseline clearly demonstrates the importance of introducing uncertainty based adaptive margin in the cross-entropy loss. We confirm in Appendix C that ORCA's improvements are retained when the number of labeled examples of seen classes is reduced to only 10%. Additional ablation studies on the importance of each component in the objective fumction, sensitivity analysis to parameters $\eta_1$ and $\eta_2$, and the performance with unbalanced data distributions are reported in Appendix C.

Table 1: Mean accuracy and NMI on CIFAR-10 and CIFAR-100 datasets calculated over three runs. NMI is reported only for novel classes. We use 50% classes as seen, and 50% classes as novel.

| Dataset | CIFAR-10 | | | | CIFAR-100 | | | |
|---|---|---|---|---|---|---|---|---|
| Classes | Seen | Novel | Novel (NMI) | All | Seen | Novel | Novel (NMI) | All |
| **Pseudo-labeling** | 82.3 | - | - | - | 59.8 | - | - | - |
| **DS$^3$L** | 87.4 | - | - | - | 64.3 | - | - | - |
| **DTC** | 53.9* | 39.5 | 38.6 | 38.3 | 31.3* | 22.9 | 36.6 | 18.3 |
| **RankStats** | 86.6* | 81.0 | 69.7 | 82.9 | 36.4* | 28.4 | 40.2 | 23.1 |
| **Baseline** | 87.6 | 86.6 | 77.3 | 86.9 | 55.2 | 32.0 | 46.6 | 34.8 |
| **ORCA** | **88.2** | **90.4** | **81.1** | **89.7** | **66.9** | **43.0** | **52.1** | **48.1** |

Table 2: Mean accuracy and NMI on ImageNet-100 calculated over three runs. NMI is reported only for novel classes.

| Split | 50 seen, 50 novel | | | | 25 seen, 75 novel | | | |
|---|---|---|---|---|---|---|---|---|
| Classes | Seen | Novel | Novel (NMI) | All | Seen | Novel | Novel (NMI) | All |
| **Pseudo-labeling** | 77.1 | - | - | - | 76.4 | - | - | - |
| **DS$^3$L** | 83.5 | - | - | - | 86.7 | - | - | - |
| **DTC** | 25.6* | 20.8 | 31.6 | 21.3 | 23.5* | 18.1 | 26.4 | 16.2 |
| **RankStats** | 47.3* | 28.7 | 43.5 | 40.3 | 34.3* | 27.8 | 39.5 | 29.8 |
| **Baseline** | 80.4 | 43.7 | 53.9 | 55.1 | 85.3 | 30.1 | 45.4 | 32.7 |
| **ORCA** | **89.1** | **72.1** | **72.5** | **77.8** | **89.4** | **67.4** | **70.2** | **69.8** |

**Evaluation with the unknown number of novel classes.** ORCA and other baselines assume that number of novel classes is known. However, in the real-world setting we often do not know number of classes in advance. In such case, we can apply ORCA by first estimating the number of classes. To evaluate performance on the CIFAR-100 dataset in this scenario, we estimate the number of clusters using technique proposed in DTC (Han et al., 2019) to be 124. We then use the estimated number of classes to re-test all the algorithms. Results shown in Table 3 show that ORCA outperforms all baselines by a large margin even when the number of novel classes needs to be estimated. Further, with the estimated number of classes ORCA achieves only slightly worse results compared to the setting in which the number of classes is known a priori.

Table 3: Mean accuracy and NMI on CIFAR-100 dataset calculated over three runs with unknown number of novel classes. We use $50\%$, $50\%$ split for seen and novel classes here.

| Classes | Seen | Novel | Novel (NMI) | All |
|---|---|---|---|---|
| **DTC** | 30.7* | 15.4 | 33.7 | 14.5 |
| **RankStats** | 33.7* | 22.1 | 37.4 | 20.3 |
| **Baseline** | 53.2 | 30.2 | 45.0 | 31.1 |
| **ORCA** | 66.3 | 40.0 | 50.9 | 46.4 |

**Effect of number of novel classes.** We systematically evaluate performance when varying the ratio of seen and novel classes in the unlabeled set on the CIFAR-100 dataset (Figure 2). We find that ORCA consistently achieves highest accuracy on both seen and novel classes across all values. The only exception is DS$^3$L approach on seen classes when the number of seen classes is very low. On seen classes, ORCA is the only method that retains stable performance across varying ratio of seen and novel classes. In contrast, pseudo-labelling SSL method significantly degrades performance with the large number of seen classes, while DS$^3$L degrades performance when seen and novel classes are equally distributed. On unseen classes, ORCA consistently outperforms baseline and novel class discovery methods by a large margin.

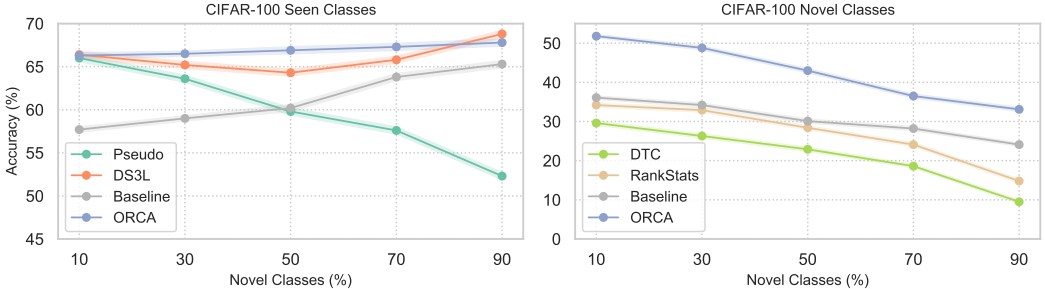

Figure 2: Average accuracy on recognizing seen classes (left) and discovering novel classes (right) when varying percentage of seen/novel classes on the CIFAR-100 dataset..

**Benefits of uncertainty based adaptive margin.** We evaluate the effect of introducing uncertainty based adaptive margin in the cross-entropy loss of ORCA. The results on the CIFAR-100 dataset are reported in Figure 3. We compare ORCA to baseline approach with zero margin, as well as to the fixed negative margin with the value of margin set to 0.5. During training, we report accuracy and uncertainty which captures intra-class variance, defined in equation (4). We find that the baseline approach is not able to reduce intra-class variance on novel class during training, resulting in the degraded performance on a novel class discovery task. In contrast, ORCA effectively reduces intra-class variance on both seen and novel classes. On seen classes, baseline reaches high performance very quickly; however, its accuracy starts to decrease close to the end of training. On the other hand, ORCA improves accuracy as training proceeds. This finding is in line with the idea that we need to slowly increase discriminability of the model as proposed in ORCA. While fixed and adaptive margin show similar intra-class variance and accuracy on novel classes, adaptive margin shows clear benefits on seen classes, achieving lower intra-class variance and significantly outperforming fixed margin during the whole training process. At the end of training, adaptive margin used in ORCA improves performance over zero margin used in baseline by 21% on seen and 34% on novel

classes (Table 1). The benefits of adaptive margin are also visible when directly comparing the quality of generated pseudo-labels (Appendix C). Taken together, these results strongly support the importance of the uncertainty based adaptive negative margin. We inspect the effect of uncertainty strength parameter $\lambda$ in Appendix C.

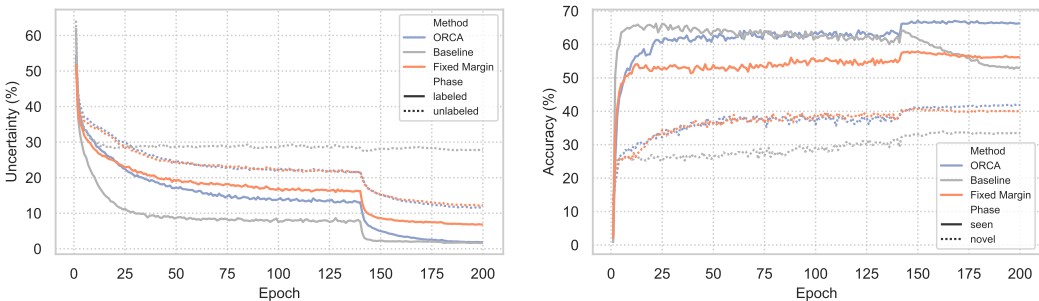

Figure 3: Effect of the uncertainty based adaptive margin on the estimated uncertainty (left) and accuracy (right) during training on the CIFAR-100 dataset.

## 5 CONCLUSION

We introduced open-world semi-supervised learning (SSL) setting in which the methods need an ability to recognize classes previously encountered in the labeled dataset, as well as discovering novel, never-before-seen classes. To address this problem, we proposed ORCA, an open-world SSL method that effectively trades off intra-class variance with uncertainty based adaptive margin. We showed that ORCA significantly outperforms SSL baselines on the task of recognizing seen classes and novel class discovery baselines on clustering unseen classes. ORCA is a unique method that can jointly solve both tasks of open-world SSL. Our work makes an important step towards designing methods for the more realistic open-world setting.

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

# A  ALGORITHM

We summarize ORCA algorithm in Algorithm 1.

---

**Algorithm 1** ORCA: **O**pen-wo**R**ld with un**C**ertainty based **A**daptive margin

---

**Require:** Labeled subset $\mathcal{D}_l = \{(x_i, y_i)\}_{i=1}^m$, unlabeled subset $\mathcal{D}_u = \{(x_i)\}_{i=1}^n$, number of novel classes, a parameterized backbone $f_\theta$, linear classifier with weight $W$.

1: Pretrain the model parameters $\theta$ with pretext loss
2: **for** epoch $= 1$ to $E$ **do**
3:     $\bar{u} \leftarrow$ EstimateUncertainty$(\mathcal{D}_u)$
4:     **for** $t = 1$ to $T$ **do**
5:         $\mathcal{X}_l,\ \mathcal{X}_u \leftarrow$ SampleMiniBatch$(\mathcal{D}_l \cup \mathcal{D}_u)$
6:         $\mathcal{Z}_l,\ \mathcal{Z}_u \leftarrow$ Forward$(\mathcal{X}_l \cup \mathcal{X}_u; f_\theta)$
7:         $\mathcal{Z}'_l,\ \mathcal{Z}'_u \leftarrow$ FindClosest$(\mathcal{Z}_l \cup \mathcal{Z}_u)$
8:         $\mathcal{L}_{\text{BCE}} \leftarrow \frac{1}{m+n} \sum_{z_i, z'_i \in (\mathcal{Z}_l \cup \mathcal{Z}_u, \mathcal{Z}'_l \cup \mathcal{Z}'_u)} -\log\langle \text{softmax}(sW^T \cdot z_i), \text{softmax}(sW^T \cdot z'_i)\rangle$
9:         $\mathcal{L}_{\text{CE}} \leftarrow \frac{1}{m} \sum_{z_i \in \mathcal{Z}_l} -\log \frac{e^{s(W_{y_i}^T \cdot z_i + \lambda \bar{u})}}{e^{s(W_{y_i}^T \cdot z_i + \lambda \bar{u})} + \sum_{j \neq i} e^{sW_{y_j}^T \cdot z_i}}$
10:         $\mathcal{L}_{\text{R}} \leftarrow KL(\frac{1}{m+n} \sum_{z_i \in \mathcal{Z}_l \cup \mathcal{Z}_u} \text{softmax}(sW^T \cdot z_i) \| \mathcal{U}(y))$
11:
12:         $f_\theta \leftarrow$ SGD with loss $\mathcal{L}_{\text{BCE}} + \eta_1 \mathcal{L}_{\text{CE}} + \eta_2 \mathcal{L}_{\text{R}}$

---

# B  IMPLEMENTATION DETAILS

Our core algorithm is developed using PyTorch (Paszke et al., 2019) and we conduct all the experiments with NVIDIA RTX 2080 Ti.

**Implementation details for CIFAR.** We follow the simple data augmentation suggested in  (He et al., 2016) with only random crop and horizontal flip. We use a modified ResNet-18 that is compatible with input size $32 \times 32$ following (Han et al., 2020) and repeat all experiments for 3 runs. We train the model using standard SGD with momentum of 0.9, weight decay of $5 \times 10^{-4}$. The model is trained for 200 epochs with a batch size of 512. We anneal the learning rate by a factor of 10 at epoch 140 and 180. Similar to Han et al. (2020), we only update the parameters of the last block of ResNet in the second training stage to avoid over-fitting. We set hyperparameters to the following default values: $s = 10$, $\lambda = 1$, $\eta_1 = 1$, $\eta_2 = 1$. They remain the same across all experiments unless otherwise specified.

**Implementation details for ImageNet.** We follow the standard data augmentation including random resized crop and horizontal flip (He et al., 2016). We use ResNet-50 as backbone. We train the model using standard SGD with momentum of 0.9, weight decay of $1 \times 10^{-4}$. The model is trained for 90 epochs with a batch size of 512. We anneal the learning rate by a factor of 10 at epoch 30 and 60. Similar to Han et al. (2020), we only update the parameters of the last block of ResNet in the second training stage to avoid over-fitting. We set hyperparameters to the following default values: $s = 10$, $\lambda = 1$, $\eta_1 = 1$, $\eta_2 = 1$. They remain the same across all experiments unless otherwise specified.

# C  ADDITIONAL RESULTS

**Results with less labeled data.** We compare performance of ORCA and baselines with the reduced number of labeled data. Instead of constructing labeled set with $50\%$ examples labeled in seen classes, we evaluate the performance on the labeled set with only $10\%$ labeled examples. Results are shown in Table 4. We find that ORCA's substantial improvements over SSL methods on seen classes and novel class discovery methods on novel classes are retained. Specifically, ORCA achieves $7.3\%$ and $32.2\%$ improvements over DS$^3$L on seen classes of CIFAR-10 and CIFAR-100 datasets, respectively. On novel classes, ORCA achieves $33.8\%$ and $90.4\%$ improvements over RankStats on CIFAR-10 and CIFAR-100 datasets, respectively.

Table 4: Mean accuracy and NMI on CIFAR-10 and CIFAR-100 datasets calculated over three runs. NMI is reported only for novel classes. For each seen class we only label $10\%$ of the examples of seen classes.

| Dataset | CIFAR-10 | | | | CIFAR-100 | | | |
|---|---|---|---|---|---|---|---|---|
| Classes | Seen | Novel | Novel (NMI) | All | Seen | Novel | Novel (NMI) | All |
| Pseudo-labeling | 67.4 | - | - | - | 10.9 | - | - | - |
| DS$^3$L | 77.2 | - | - | - | 39.7 | - | - | - |
| DTC | 42.7* | 31.8 | 33.5 | 32.4 | 22.1* | 10.5 | 23.5 | 13.7 |
| RankStats | 71.4* | 63.9 | 60.5 | 66.7 | 20.4* | 16.7 | 32.5 | 17.8 |
| Baseline | 82.7 | 70.6 | 67.5 | 72.4 | 35.8 | 23.9 | 36.4 | 22.2 |
| ORCA | **82.8** | **85.5** | **73.5** | **84.1** | **52.5** | **31.8** | **44.8** | **38.6** |

**Ablation study on the objective function.** The objective function in the proposed open-world SSL baseline and ORCA consists of supervised loss, pairwise loss, and regularization towards uniform distribution. To investigate importance of each part, we conduct an ablation study in which we modify baseline approach by removing: (i) supervised loss, (ii) regularization towards uniform distribution. In the first case, we rely only on the regularized pairwise loss to solve the problem, while in the second case we have unregularized supervised and pairwise losses. We note that the pairwise loss is essential in order to be able to discover novel classes. We show results on CIFAR-10 and CIFAR-100 datasets in Table 5. Results demonstrate that both supervised loss and regularization are essential parts of the designed objective function and significantly improve performance of the open-world SSL baseline, and consequently ORCA.

Table 5: Ablation study on components of the objective function. We report mean accuracy and NMI over three runs. We use $50\%$, $50\%$ split for seen and novel classes.

| Dataset | CIFAR-10 | | | | CIFAR-100 | | | |
|---|---|---|---|---|---|---|---|---|
| Classes | Seen | Novel | Novel (NMI) | All | Seen | Novel | Novel (NMI) | All |
| w/o $\mathcal{L}_{\text{CE}}$ | 70.6 | 78.9 | 73.4 | 76.2 | 12.2 | 13.4 | 25.4 | 10.0 |
| w/o $\mathcal{L}_{\text{R}}$ | 90.2 | 47.9 | 68.3 | 53.4 | 51.2 | 16.1 | 33.7 | 20.4 |
| Baseline | 87.6 | 86.6 | 77.3 | 86.9 | 55.2 | 32.0 | 46.6 | 34.8 |

**Sensitivity analysis of $\eta_1$ and $\eta_2$.** Parameters $\eta_1$ and $\eta_2$ define importance of the supervised loss and regularization towards uniform distribution, respectively. To analyze their effect on the performance, we vary these parameters and evaluate the ORCA's performance on the CIFAR-100 dataset. We find that higher values of $\eta_1$ achieve slightly better performance on seen classes. This result agrees well with the intuition: giving more importance to the supervised loss improves performance on the seen classes. The effect of parameter $\eta_2$ on seen classes is opposite and lower values of $\eta_2$ achieve better performance on seen classes. On novel classes, the optimal performance is obtained when $\eta_1$ and $\eta_2$ are set to 1.

Table 6: Mean accuracy and NMI computed over three runs with different values of $\eta_1$ and $\eta_2$ on the CIFAR-100 dataset with 50%, 50% split for seen and novel classes.

| $\eta_1$ | Seen | Novel | Novel (NMI) | All | $\eta_2$ | Seen | Novel | Novel (NMI) | All |
|---|---|---|---|---|---|---|---|---|---|
| 0.6 | 65.7 | 42.3 | 51.3 | 47.5 | 0.6 | 71.4 | 28.0 | 45.5 | 30.2 |
| 0.8 | 66.0 | 41.9 | 50.9 | 47.1 | 0.8 | 68.5 | 39.3 | 50.5 | 43.0 |
| 1.0 | 66.9 | 43.0 | 52.1 | 48.1 | 1.0 | 66.9 | 43.0 | 52.1 | 48.1 |
| 1.2 | 66.9 | 42.7 | 51.3 | 47.6 | 1.2 | 66.7 | 42.8 | 51.7 | 47.9 |
| 1.4 | 66.6 | 41.9 | 50.4 | 46.8 | 1.4 | 66.3 | 41.8 | 50.9 | 47.7 |

**Sensitivity analysis of uncertainty regularizer $\lambda$.** The intention of introducing the uncertainty based adaptive margin is to enforce the group of labeled and unlabeled data to have similar intra-class variances. Here we inspect how does the uncertainty regularizer $\lambda$ affect performance. The results are shown in Table 7. A slightly larger $\lambda$ achieves higher accuracy on the novel classes with

the cost of lower accuracy on seen classes. In contrast, smaller values of $\lambda$ achieve slightly better performance on seen classes. In general, ORCA is robust to the selection of the $\lambda$ parameter.

Table 7: Mean accuracy with different values of regularizer $\lambda$ on CIFAR-100 dataset with 50%, 50% split for seen and novel classes.

| $\lambda$ | Seen | Novel | Novel (NMI) | All |
|---|---|---|---|---|
| 0.6 | 67.0 | 42.6 | 51.6 | 47.5 |
| 0.8 | 67.0 | 42.8 | 51.6 | 47.6 |
| 1.0 | 66.9 | 43.0 | 52.1 | 48.1 |
| 1.2 | 66.6 | 43.4 | 52.0 | 48.0 |
| 1.4 | 66.0 | 43.5 | 52.2 | 48.2 |

**Ablation study on unbalanced data distribution.** To check whether proposed regularization towards uniform distribution negatively affects performance when the distribution of the classes is unbalanced, we artificially introduce unbalanced distributions in CIFAR-10 and CIFAR-100 datasets. In particular, we make distributions long-tailed by following an exponential decay in sample sizes across different classes. The imbalance ratio between sample sizes of the most frequent and least frequent class is set to 10 in the experiments. The results are shown in Table 8. On the unbalanced CIFAR-10 dataset, proposed regularization improves accuracy by 7.4% on seen classes and by 35% on novel classes over the non-regularized model. On the unbalanced CIFAR-100 dataset, regularization improves the performance by 16.9% on seen and 5.1% on novel classes. These results demonstrate the advantage of the proposed regularization even when the classes are unbalanced.

Table 8: Mean accuracy and NMI on unbalanced CIFAR-10 and CIFAR-100 datasets calculated over three runs.

| Dataset | | CIFAR-10 | | | | CIFAR-100 | | | |
|---|---|---|---|---|---|---|---|---|---|
| Classes | Seen | Novel | Novel (NMI) | All | Seen | Novel | Novel (NMI) | All |
| w/o $\mathcal{L}_R$ | 84.2 | 61.4 | 64.6 | 62.8 | 55.6 | 35.4 | 50.6 | 35.2 |
| w/ $\mathcal{L}_R$ | 90.4 | 82.9 | 74.6 | 69.0 | 65.0 | 37.2 | 53.8 | 40.5 |

**Benefits of uncertainty based adaptive margin on pseudo-labels accuracy.** The benefit of the uncertainty based adaptive margin is that it reduces the bias towards seen classes. To evaluate the effect of uncertainty based adaptive margin on the quality of generated pseudo-labels during training, we compare the accuracy of adaptive margin to baseline approach with zero margin and fixed negative margin adaptation on the CIFAR-100 dataset. We report accuracy of generated pseudo-labels in Figure 4, following the same setting as in Figure 3. This analysis additionally confirms that adaptive margin increases the accuracy of the estimated pseudo-labels.

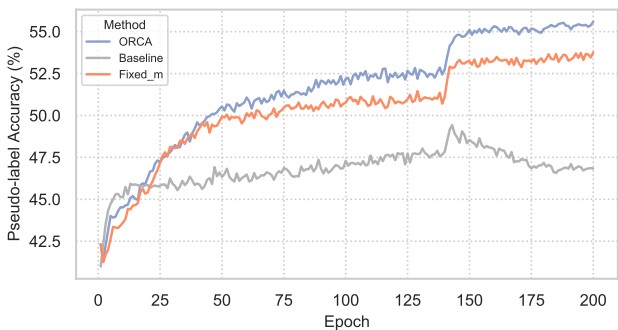

Figure 4: Effect of the uncertainty based adaptive margin on the quality of estimated pseudo-labels during training on the CIFAR-100 dataset.

