# OpenReview forum: "Open-world Semi-supervised Learning"
_ICLR.cc/2021/Conference — Reject_

### Official Review · AnonReviewer3 · 2020-10-27
**OPEN-WORLD SEMI-SUPERVISED LEARNING**

**Rating:** 6
**Confidence:** 3

**Review:**

Summary: the authors propose Open World Semi-Supervised Learning (ORCA), a semi-supervised method that learns to classify previously seen classes in the labeled data and novel class in the unlabeled data. The proposed method is end to end and achieves significant improvement on ImageNet dataset compared to baseline methods.

Pros:
The paper takes one of the most important issue of semi-supervised learning: recognizing novel classes. For me, the problem itself is real and practical.

The experiments are strong and methods shows significant improvement compared to competing methods.

Cons:
The paper lacks clarity in some sections. The writing could be better and the notations better defined.

Comments:
The unsupervised loss function is not clear to me, I find it difficult to parse equation 1 from the way it's written. Could you please explain the equation better?

It's my understanding that each z_i in the unlabeled set is assigned a class based on the ranking of pairwise distance in the mini batch.  If the mini batch of z_i contains labeled samples, then it could be assigned to the labels example if the images are similar. For example if there's an unlabeled example of a horse, it could be assigned to the labeled donkey class. Wouldn't this be problematic for the novel class prediction if the batches always contain similar examples from different classes?

Why the choice of cosine distance for the similarity measure? Using cosine similarity means that you do not consider the magnitude of the embeddings. This would make sense for text embeddings but the experiments are on image datasets

---

> ### Author Response · Authors · 2020-11-17
> **Responses to R3**
>
> We thank the reviewer for the positive evaluation of our work and for recognizing the problem addressed in our work as one of the most important issues of semi-supervised learning.
>
> *RE: The unsupervised loss function is not clear to me, I find it difficult to parse equation 1 from the way it's written. Could you please explain the equation better?*
>
> The reviewer is correct that we rank pairwise distances in the mini-batch. Pseudo-labels are then generated for each datapoint based only on its most similar neighbor. Our pairwise loss is the binary cross entropy between example and its most similar neighbor. We only consider positive pairs, since we find that negative pairs (i.e. pseudo-labels based on confident negative examples) do not contribute to learning. The reason is that negative examples are very easily recognized. For labeled examples, we have the ground truth annotations, so we use them to compute the loss. For unlabeled examples, we compute loss based on the generated pseudo-labels. Since our batch size is much larger than the number of classes, it is highly unlikely that two data points from the same class would not be in the same batch. In the example provided by the reviewer, if we are looking at the most similar neighbor of the horse unlabeled example and there is another horse image in the batch it should be closer to the target horse example than the donkey example. We have now changed the name of unsupervised loss to pairwise loss since we also use ground truth annotations from the labeled data to compute it. We hope this will help in clarification. We also devoted more space to explain pairwise loss better as suggested by the reviewer.
>
> *RE: Why the choice of cosine distance for the similarity measure?*
>
> We thank the reviewer for this important question. We use cosine similarity since we do not take magnitudes into consideration. The reason is that magnitudes would be larger for seen classes and lead to a bias towards seen classes. We note that cosine distance has been used on many image datasets, e.g., [1-3] to name a few.
>
> [1] Chang, et al. Deep Adaptive Image Clustering. ICCV ‘17
> [2] Vinyals, et al. Matching networks for one shot learning. NeurIPS ‘16
> [3] Peng, et al. Few-Shot Image Recognition with Knowledge Transfer. ICCV ‘19

---

### Official Review · AnonReviewer2 · 2020-10-27
**This study is well-motivated and tackles an important problem, but some points remain unclear.**

**Rating:** 6
**Confidence:** 4

**Review:**

--Paper summary--

The authors propose a method to tackle a new problem setting of semi-supervised learning, called an open-world semi-supervised learning, where the model is required to accurately discriminate known-class data as well as to appropriately discover unknown classes contained in an unlabeled dataset. The objective function to be minimized in the proposed method comprises three terms: unsupervised loss, supervised loss with uncertainty based adaptive margin, and entropy regularization. The experimental results with several datasets have validated the advantage of the proposed method.

--Review summary--

This study is well-motivated and tackles an important problem that would occur in real-world applications. The design of the proposed method seems reasonable, and it works well in the experiments with several datasets. However, some points remain unclear or are not convincing, which makes my score a bit conservative. To appropriately determine my score, I would appreciate if the authors clarify these points in their response.

--Details--

Strength

- The setting of the open-world semi-supervised learning is interesting and should be practically important.
- The paper is well-written and well-organized.

Weakness and concerns

- It is not clear whether the design of the proposed method really leads to the improvement of the performance or not. Since the authors use SimCLR to pretrain the model, the pretrained model has already had a substantially good feature representation. Consequently, if we know the number of classes in the training data, unsupervised learning methods like [R1] (or simple clustering methods) might work well to discriminate all classes. Although the authors adopt a new idea to the supervised loss, it seems unclear whether the supervised loss itself really contributes to good performance. Are there any experimental result on performance sensitivity to \eta?
[R1] "Learning Discrete Representations via Information Maximizing Self-Augmented Training," ICML 2017.
- The experimental setup is not convincing. Due to the motivation of SSL, the number of unlabeled data should be much larger than that of labeled data, and it is often tens or hundreds times larger in the literature. However, in this paper, it is basically 3~7 times larger in the experiments. Is there any reason why the authors did not much reduce the number of labeled data?
- Two concerns on uncertainty based adaptive margin.
	- How did the authors estimate the class posterior probability in Eq. (4)? Is it just the output of the softmax function?
	- Since this adaptive margin is adopted to improve the accuracy of pseudo-labels, how much it is improved should be reported in Fig. 3.
- How did the authors conduct validation? Due to the existence of unseen classes in unlabeled data, how to conduct validation is not trivial.
- Is it reasonable to call L_BCE unsupervised loss? Since Z_l' is obtained using ground-truth labels, L_BCE cannot be computed in an unsupervised manner.

Minor concerns
- I could not get the reason why the proposed method is called ORCA.


--After receiving authors' response--

I would like to thank the authors for providing additional experimental results and giving clarifications. Since my concerns are almost solved, and I updated my score from 5 to 6. This study would be a great first step to tackle the challenging problem, open-world semi-supervised learning, though there are several remaining issues (e.g., validation is hard to conduct, the number of unknown classes should be known, etc.). I vote for "weak accept."

---

> ### Author Response · Authors · 2020-11-17
> **Responses to R2**
>
> We thank the reviewer for the comments and we highly appreciate the willingness to adjust the score based on our response. We have conducted additional experiments that further validate our method and we hope that the new results will help in increasing the reviewer's confidence.
>
> *RE: Unsupervised learning methods like [R1] (or simple clustering methods) might work well to discriminate all classes. Although the authors adopt a new idea to the supervised loss, it seems unclear whether the supervised loss itself really contributes to good performance.*
>
> We thank the reviewer for this question. Previous works on novel class discovery methods represented by DTC and RankStats methods in our evaluation have shown consistently better than clustering methods. Our experiments show that incorporating adaptive margin in the supervised loss leads to significantly better performance compared to the designed open-world SSL baseline, indicating that the supervised loss is indeed essential. To further evaluate the benefits of supervised loss, we have conducted an additional ablation study in which we completely remove supervised loss and rely only on the binary cross entropy loss for both seen and unseen classes. Results clearly demonstrate that supervised loss is essential. For instance, on CIFAR-100 dataset accuracy of the designed open-world SSL baseline is improved by 352% on seen classes, and by 138% on novel classes by incorporating supervised loss. Improvements in ORCA are even higher. These results clearly show benefits of the supervised loss. In response to the reviewer’s feedback, we included these results in Table 5 in Appendix C.
>
> *RE: Are there any experimental result on performance sensitivity to \eta?*
>
> We thank the reviewer for this question. We have conducted additional experiments on the sensitivity to \eta. We find that higher values of \eta_1 achieve slightly better performance on seen classes. This result agrees well with the intuition: giving more importance to the supervised loss improves performance on the seen classes. The effect of parameter \eta_2 on seen classes is opposite and lower values of \eta_2 achieve better performance on seen classes. On novel classes, the optimal performance is obtained when \eta_1 and \eta_2 are set to 1. We have included this new ablation study in Table 6 in Appendix C.
>
> *RE: Is there any reason why the authors did not much reduce the number of labeled data?*
>
> We thank the reviewer for this important question. We have now tested ORCA by labeling only 10% examples of seen classes.  We find that ORCA's substantial improvements over SSL methods on seen classes and novel class discovery methods on novel classes are retained. In particular, on the CIFAR-10 dataset ORCA achieves 7.3% relative improvement on the seen classes over SSL methods, and 33.8% improvement on the novel classes over methods for novel class discovery. On the CIFAR-100 dataset, ORCA achieves 32.2% and 90.4% improvements on seen and novel classes, respectively. We have added these results in Table 4 in Appendix C.
>
> *RE: How did the authors estimate the class posterior probability in Eq. (4)?*
>
> The reviewer is correct that the class posterior probability is estimated as the output of the softmax. We have added the explanation in the paper as suggested by the reviewer.
>
> *RE: How much the adaptive margin improves accuracy should be reported in Fig 3.*
>
> We thank the reviewer for the comment. The final  improvement of adaptive margin used in ORCA over baseline approach without adaptive margin is reported in Table 1. We have now additionally added the improvements in the paragraph devoted to Fig 3 as suggested by the reviewer.
>
> *RE: How did the authors conduct validation?*
>
> We thank the reviewer for the question. For all the experiments, we simply take the model trained after the last epoch to test the performance on unlabeled data. As it can be verified from Figure 3, the performance of our proposed ORCA model is quite stable at the end of the training.
>
> *RE: Is it reasonable to call L_BCE unsupervised loss?*
>
> We thank the reviewer for this very insightful suggestion. We agree that the name may be misleading since we also use ground truth annotations for computing the loss. We have changed the terminology and we are now referring to this loss as pairwise loss instead of unsupervised loss.
>
> *RE: I could not get the reason why the proposed method is called ORCA.*
>
> We thank the reviewer for this question. ORCA stands for Open-woRld with unCertainty based Adaptive margin. We have explained the used acronym as suggested by the reviewer.

---

> > ### Comment · AnonReviewer2 · 2020-11-18
> > **Thank you for your response and providing additional experimental results.**
> >
> > Thank you for your response and providing additional experimental results. They greatly help my understanding.
> >
> > Let me confirm two more things.
> >
> > > RE: How much the adaptive margin improves accuracy should be reported in Fig 3.
> >
> > I thought that Fig. 3 shows test accuracy. Is it correct? What I'm interested in is accuracy of pseudo-labels, that is actually training accuracy, and how it is improved by adopting the adaptive margin.
> >
> > > RE: How did the authors conduct validation?
> >
> > Let me ask it in another way. How did you tune \eta or other hyper-parameters? Even when we have validation dataset, it is essentially hard to evaluate the performance of the trained model due to the existence of unseen classes.

---

> > > ### Author Response · Authors · 2020-11-20
> > > **Response to R2**
> > >
> > > We thank the reviewer for the prompt response and follow-up questions.
> > >
> > > *I thought that Fig. 3 shows test accuracy. Is it correct? What I'm interested in is accuracy of pseudo-labels, that is actually training accuracy, and how it is improved by adopting the adaptive margin.*
> > >
> > > We thank the reviewer for the question. We misunderstood the question first time, it is correct that Figure 3 shows the test accuracy. In response to the reviewer’s question, we have now directly tested the accuracy of pseudo-labels. Similar to Figure 3, the results show improvements in the accuracy of pseudo-labels obtained with the adaptive margin compared to the baseline approach and fixed margin. We have included these results as Figure 4 in Appendix C. We thank the reviewer for suggesting this experiment.
> > >
> > > *Let me ask it in another way. How did you tune \eta or other hyper-parameters? Even when we have validation dataset, it is essentially hard to evaluate the performance of the trained model due to the existence of unseen classes.*
> > >
> > > We thank the reviewer for the question. We agree it is hard to evaluate performance on the separate validation set due to the presence of unseen classes. We did not tune hyperparameters. Specifically, in our method the only hyperparameters that are method-specific are \eta and \lambda. We set them to default values of 1 and we did not tune them further. We check the robustness to these parameters and report them in the Tables 6 and 7 in Appendix C. All other hyperparameters are set to default values and they have not been tuned. Therefore, all hyperparameters are shared between all three datasets. We note that it would be possible to construct a validation set by selecting part of seen classes as unseen validation classes but this would complicate the evaluation process.

---

### Official Review · AnonReviewer4 · 2020-10-29
**The paper proposed a novel method for solving open-world semi-supervised learning problem.**

**Rating:** 6
**Confidence:** 4

**Review:**

Pros
- The paper explores an interesting semi-supervised learning (SSL) setting in which the unlabeled data contain not only the seen class but also novel classes. The problem is interesting in that it is more practical than the classic SSL setting in the real world and has been seldomly researched.
- The method includes a self-supervised pretraining step and a finetune step. The finetune step jointly solves the classification task and the clustering task with a unified objective loss. The supervised loss, which is the main contribution of the method, overcomes the imbalance problem caused by BCE with a novel adaptive margin-based loss. The authors validate this loss with empirical results.
- The paper is well-organized and clearly written. As far as I can see, the method is technically sound.

Cons
- The comparison methods (such as pseudo-labeling, DS3L) are not strong enough. It is suggested to compare with more SOTA methods. Since SimCLR is more powerful than RotNet, it is strange to see RankStats get worse performance than the original paper.
- The number of novel classes has to be prefixed.

---

> ### Author Response · Authors · 2020-11-17
> **Responses to R4**
>
> We thank the reviewer for the valuable feedback and positive evaluation of our work. We are glad to hear that the reviewer finds our setting interesting and practical, and method technically  sound.
>
> *RE: The comparison methods (such as pseudo-labeling, DS3L) are not strong enough. It is suggested to compare with more SOTA methods*
>
> We thank the reviewer for the suggestion. However, DS3L is the SOTA method published at ICML ’20 and shown to consistently outperform existing SSL methods in the setting of unseen-class unlabeled data. While the reference of DS3L in the related work is correct, we realized that we wrongly referred to the older paper by Guo in the baselines paragraph of the experimental setting (‘17 instead of ‘20). We believe that this is the reason for the misunderstanding and we thank the reviewer for pointing this out. We have fixed the typo.
>
> *RE:  RankStats get worse performance than the original paper*
>
> SimCLR is indeed more powerful than RotNet. The reason why RankStats has worse performance than the original paper is that our evaluation setting is much more difficult than the setting of the RankStats paper. First, RankStats paper assumes all examples of the seen classes are labeled, while we label only 50% samples of the seen classes, and use the rest as unlabeled set. Therefore, in our setting RankStats method needs to be able to recognize seen classes and identify novel classes, which is a harder task than only discovering novel classes as it is the case in the RankStats paper. For instance, on the CIFAR-10 dataset this means that in our setting the task is 10-class classification, compared to 5-class classification in RankStats. Secondly, on the CIFAR-100 and ImageNet datasets the additional level of difficulty in our setting is that we use a larger number of unseen classes. On CIFAR-100 we evaluate the methods with 50 unseen classes, and on ImageNet with 50 and 75of unseen classes. In contrast, RankStats evaluates the approach with 10 and 30 unseen classes on CIFAR-100 and ImageNet datasets, respectively.
>
> *RE: The number of novel classes has to be prefixed*
>
> We thank the reviewer for this insightful comment. We addressed this point in Appendix C. In particular, we showed that if the number of classes is not known in advance, we can estimate the number of classes and ORCA’s performance is only slightly degraded compared to the setting in which we assume that the number of classes is known. Further, ORCA significantly outperforms baseline methods even with the estimated number of classes. In response to the reviewer’s feedback, we have now moved these results to the main paper (now Table 3) to emphasize this point.

---

### Official Review · AnonReviewer1 · 2020-10-29
**Open-world Semi-supervised Learning**

**Rating:** 6
**Confidence:** 4

**Review:**

The paper considers open-world SSL settings  where the model recognizes previously seen classes, and detects novel classes which are not present in the labeled dataset. The method contains three losses to train a model in this setting: a) supervised loss on labeled data, b) unsupervised loss on unlabeled data from pseudo-labels obtained  from confident pairwise similarities, and c)  regularization term that avoids assigning all the unlabeled samples to the same class. Then the paper evaluates the effectiveness of the  method on CIFAR-10/100 and ImageNet-100 datasets.

The paper is well organized, well-written, and tackle SSL problem in a more realistic setting. Experiments are enough to some extent. However, I feel that the paper just combines several well-known and well-studied techniques for different tasks which intersect with open-world SSL settings. (e.g., self-supervised learning in SSL, clustering for SSL or clustering for transferring knowledge  across domains and tasks, using confident pseudo-labels for training an SSL model, etc). From this perspective, technical novelty of the work is limited, although experiments show good results for open-world SSL setting.


-First, I think this set-up does not contain all the scenarios for a real-world SSL. For example,  this set-up does not consider covariate shift where the data belonging to the same classes but different image statistics or style (e.g., dogs in natural images, dogs in the painting or sketches images). However, in the real-world scenario, we may have images of the same class but different domains.


-The paper adds several losses studied in the literature without analyzing if any has a negative effect on the others in the open-world SSL setting.

-Assuming that the number of novel classes |C_u| is known is a bit unrealistic to me in the open-world SSL setting.

-The method ranks the distances, and for each sample generates the pseudo-label for its most similar neighbor. However,  in an open world SSL setting due to the very-limited-label regime, the representation may not be ideal, and  therefore, pairwise-labeling may not be ideal and can possibly propagate the error through the network over the course of training.


-Does the regularization towards uniform distribution consider unbalanced novel classes which is common in open-world SSL?

-Does the method perform well in cases where the test set contains novel classes that do not appear in the unlabeled set?

Generally, the work has some potential from practical point of view, however, it needs more work to be improved technically.

---

> ### Author Response · Authors · 2020-11-17
> **Responses to R1 (part 2)**
>
> *RE: Does the regularization towards uniform distribution consider unbalanced novel classes which is common in open-world SSL?*
>
> We thank the reviewer for the question. We have now tested ORCA with and without proposed regularization by artificially introducing unbalanced distribution of the classes in CIFAR-10 and CIFAR-100 datasets. We make distribution long-tailed by following an exponential decay in sample sizes across different classes. On the unbalanced CIFAR-10 dataset, proposed regularization improves accuracy by 7.4% on seen classes and by 35% on unseen classes over the non-regularized model. On the unbalanced CIFAR-100 dataset, regularization improves the performance by 16.9% and 5.1% on seen and unseen classes, respectively. These results demonstrate the advantage of our regularization even when the classes are unbalanced. We have included these new results in Table 8 in Appendix C.
>
> *RE: Does the method perform well in cases where the test set contains novel classes that do not appear in the unlabeled set?*
>
> The goal of the proposed method is to recognize seen and discover novel classes. The novel class discovery task boils down to clustering so it is natural to pose it as a problem of labelling unlabeled data. It is not clear how the clustering task could be solved in the inductive setting suggested by the reviewer.

---

> ### Author Response · Authors · 2020-11-17
> **Responses to R1 (part 1)**
>
> We thank the reviewer for the valuable feedback and insightful comments. Based on the reviewer’s comments, we have conducted additional experiments that further validate our method. We believe that the results of the newly conducted experiments further strengthen our work.
>
> *RE:  From this perspective, technical novelty of the work is limited, although experiments show good results for open-world SSL setting.*
>
> We thank the reviewer for the comment. We would like to point out that our contributions are twofold. First, we introduce an open-world SSL setting and show that none of the existing methods can effectively solve the problem. This setting is more realistic and more difficult than the recently studied SSL setting in which algorithm needs to reject unseen classes. Besides this contribution, we propose a unique method for solving the open-world SSL problem. The key technical novelty in our method is in introducing uncertainty based adaptive margin. We show that naively combining losses of well-studied techniques leads to a bias towards seen classes which reduces the ability to adapt to novel classes, while adaptive margin mitigates this bias. Our experimental results strongly support the importance of the adaptive margin as a key novelty for effectively solving the challenging open-world SSL problem.
>
> *RE: This set-up does not consider covariate shift where the data belonging to the same classes but different image statistics or style.*
>
> We agree with the reviewer that it would be very beneficial to tackle the same problem in the domain adaptation setting by assuming covariate shift between the classes. However, in this work we are interested in the semi-supervised learning (SSL) setting in which there is no such assumption on the covariate shift. Therefore, assuming the covariate shift leads to a new/different problem. We plan to work on the proposed problem in our future work. We believe that the proposed open-world SSL setting is very realistic since in many domains we can label only a small amount of data but can not guarantee that we will cover all possible classes which is assumed by conventional SSL methods.
>
> *RE: The paper adds several losses studied in the literature without analyzing if any has a negative effect on the others in the open-world SSL setting.*
>
> We thank the reviewer for this question. We  have created a very strong baseline for open-world SSL task which combines supervised, unsupervised/pairwise losses and regularization towards uniform distribution. To directly test the effect of each component in the objective function, we have now conducted a new ablation study suggested by the reviewer. First, we remove supervised loss and we use the unsupervised/pairwise loss for both seen and unseen classes. Next, we remove regularization towards uniform distribution while retaining supervised and unsupervised/pairwise loss. We note that the unsupervised/pairwise loss is essential in order to be able to discover novel classes. The results show that both supervised loss and regularization towards uniform distribution are essential parts of our objective function and significantly contribute to the performance.  We included these novel results in Table 5 in Appendix C.
>
> *RE: Assuming that the number of novel classes |C_u| is known is a bit unrealistic to me in the open-world SSL setting.*
>
> We agree with the reviewer that in the realistic scenario we often do not know the number of novel classes. We addressed this point in Appendix C. In particular, we showed that if the number of classes is not known in advance,  we can estimate the number of classes and ORCA’s performance is only slightly degraded compared to the setting in which we assume that the number of classes is known. Further, ORCA significantly outperforms baseline methods even with the estimated number of classes. In response to the reviewer’s feedback, we have now moved these results to the main paper (now Table 3) to emphasize this point.
>
> *RE: pairwise-labeling may not be ideal and can possibly propagate the error through the network over the course of training.*
>
> We agree with the reviewer that pairwise labeling can possibly propagate the error through the network. To reduce the effect of wrong annotations, we pretrain the network with self-supervised learning and during fine-tuning stage only generate pseudo-labels from the most confident positive pairs in each iteration. We note that the concern of wrong annotations may be attributed to any pseudo-labelling based method. Despite this, pseudo-labeling based methods are state-of-the-art methods for semi-supervised learning.

---

### Decision · Program_Chairs · 2021-01-07
**Final Decision**

**Decision:**

Reject

**Comment:**

This work addresses a novel and important real-world setting for semi-supervised learning – the open-world problem where unlabeled data may contain novel classes that are not seen in labeled data.  The paper provides an approach by combining three loss functions: a supervised cross-entropy loss, a pairwise cross-entropy loss with adaptive uncertainty margin, and a regularization towards uniform distribution.

The authors were responsive to reviewers’ comments and have respectively improved their paper by adding experiments, including an ablation study of each component of the objective function, study of the effect regularization on unbalanced class distributions, reporting accuracy on pseudo-labels.  While two reviewers have slightly increased their scores, some concerns still remain.

This is a borderline paper, and after some discussion and calibration, we decided that the work in its current form does not quite meet the bar for acceptance.